# JudgeFlow: Agentic Workflow Optimization via Block Judge

## Abstract

Optimizing LLM-based agentic workflows is challenging for scaling AI capabilities, some current methods rely on coarse, end-to-end evaluation signals and lack fine-grained signals on where to refine, often resulting in inefficient or low-impact modifications. To address these limitations, we propose JudgeFlow, an Evaluation-Judge-Optimization-Update pipeline. We incorporate reusable and configurable logic blocks into agentic workflows, capturing fundamental forms of logic. On top of this abstraction, we design a dedicated Judge module that inspects execution traces, particularly failed runs, and assigns rank-based responsibility scores to problematic blocks. These fine-grained diagnostic signals are then leveraged by an LLM-based optimizer, which focuses modifications on the most problematic block in the workflow. Our approach improves sample efficiency, enhances interpretability through block-level diagnostics, and provides a scalable foundation for automating increasingly complex agentic workflows. We evaluate JudgeFlow on mathematical reasoning and code generation benchmarks, and the results demonstrate that JudgeFlow achieves superior performance and optimization efficiency compared to existing methods.

## 1 Introduction

Large language models (LLMs) (Brown et al., 2020) have achieved remarkable success across a wide range of domains. Moving beyond the scope of foundation models (Bommasani et al., 2022), by integrating LLMs into intelligent agent architectures, the emerging foundation agents (Liu et al., 2025a) have attracted more attention. Starting from early work on prompt engineering, such as reasoning-enhanced methods (Wei et al., 2023; Wang et al., 2023b; Yao et al., 2023a), to more recent developments in multi-agent system approaches (Du et al., 2023; Li et al., 2023; Hong et al., 2024), these handcrafted strategies have achieved strong performance across a range of tasks, including mathematical reasoning (Cobbe et al., 2021), code generation (Austin et al., 2021), and question answering (Yang et al., 2018).

However, these agentic systems still depend heavily on manual design, making workflow construction complex, costly, and inflexible. AutoML (Hutter et al., 2019) has shown that automating traditionally handcrafted and labor-intensive processes in machine learning can substantially reduce human effort and accelerate the development of high-performance models. Inspired by this success, recent efforts aim to automate the design and optimization of LLM-based agentic workflows (Lee et al., 2025). While these agentic systems still rely on LLMs as core execution engines, optimizing the LLMs themselves through pretraining or fine-tuning (Rafailov et al., 2023) often demands substantial computational resources and massive-scale data, making such approaches expensive in many settings (Kaplan et al., 2020). Instead, keeping the underlying model parameters fixed, and focusing on optimizing the systems structure and behavior leads to a more tractable and efficient optimization.

Automation efforts in agentic systems initially focused on prompt optimization, exemplified by Textual Gradients which leverage LLM feedback for end-to-end optimization (Pryzant et al., 2023; Yuksekgonul et al., 2024; Wang et al., 2024b; Yin & Wang, 2025). Current efforts are expanding to optimize architecture and execution flow of entire agentic systems. Agentic workflow can be modeled as neural network (Liu et al., 2024; Ma et al., 2025), graph (Zhuge et al., 2024a; Zhang et al., 2025a), and code (Hu et al., 2025; Zhang et al., 2025b; Zheng et al., 2025), each offering different levels of

representational capacity, interpretability, and optimization difficulty. For instance, Directed Acyclic Graphs (DAGs)-represented workflows facilitate tractable optimization but constrain the ability to represent complex structures such as loops or conditional branching. In contrast, code-represented workflows provide comprehensive expressivity in defining intricate logic and control flow, but error attribution within code execution is difficult, and optimization often has to rely solely on end-to-end evaluation signals rather than fine-grained intermediate feedback. Building on code-represented workflows, Zhang et al. (2025b) introduce operators as modular units that encapsulate common agentic actions and propose a Monte Carlo Tree Search (MCTS) framework that employs LLMs to iteratively optimize workflow structures using past experience. However, the expansion phase in MCTS and the subsequent evaluation of candidate workflows can be expensive, and the effectiveness of the optimization process is constrained by the granularity of guidance available for modifications. In the absence of sufficiently fine-grained diagnostic information to precisely identify which specific part within the complex workflow requires modification, the search may explore ineffective or low-impact alterations. Furthermore, complex code structural interactions—such as conditional constructs where only one branch of an `if-else` statement is executed along a trajectory—leave certain components without informative signals, thereby hindering fine-grained analysis.

To address these challenges, we introduce JUDGEFLOW, an Evaluation-Judge-Optimization-Update pipeline. First, we incorporate reusable and configurable logic blocks into agentic workflows. These blocks capture three fundamental forms of logic: sequential, loop, and conditional, which are able to broadly represent code-based workflows. Compared with operators, which abstract specific agentic operations or functionalities (Zhang et al., 2025b), logic blocks serve as higher-level structural abstractions. By introducing logic blocks that abstract such common control structures, we retain the structural diversity of code-represented workflows while providing an intermediate level of abstraction between operators and workflows. This additional layer facilitates interpretability and exposes more meaningful diagnostic information for subsequent optimization.

Second, we incorporate a dedicated Judge module that analyzes the execution trace, with particular emphasis on failed runs. We hypothesize that optimizers should receive both evaluation and optimization signals. For each unsuccessful execution, the Judge attempts to identify the most problematic block within the workflow as illustrated in Fig. 1. To further improve the precision of diagnosis, we adopt a rank-based approach at the block level. The resulting targeted diagnostic signals are propagated to the subsequent optimization stage, enabling more focused and efficient refinement of weak blocks. In this way, optimization efforts can be concentrated on repairing underperforming components, resulting in more effective and reliable improvements in overall workflow performance. Besides relying solely on end-to-end evaluation signals, our approach leverages block-level diagnostic information, enabling the optimizer to focus on the most problematic components.

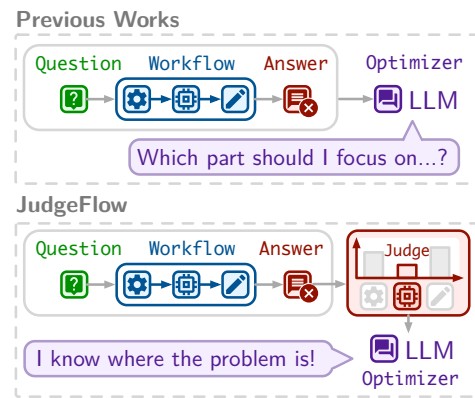

Figure 1: Block-level judging guides agentic workflow optimization by identifying the most problematic block in failed executions.

In summary, our contributions are as follows:

- We propose a novel Evaluation-Judge-Optimization-Update pipeline named JUDGEFLOW;

- We introduce reusable and configurable logic blocks as higher-level structural units, which balance the expressivity of code-based workflows with tractable optimization, while supporting interpretability and intermediate execution tracing;

- We design a Judge module that analyzes execution traces, especially failed runs, and assigns rank-based responsibility scores to problematic blocks, enabling fine-grained error localization and targeted refinement for subsequent optimization.

- We evaluate JUDGEFLOW on mathematical reasoning and code generation benchmarks, showing that it outperforms existing methods.

## 2 RELATED WORK

**LLM-based (Multi-)Agent Systems**    In recent years, LLM-based (multi-)agent systems have achieved notable successes (Wang et al., 2024a; Huang et al., 2024; Tran et al., 2025) across planning (Huang et al., 2024), reasoning (Putta et al., 2024), and human–AI coordination (Zou et al., 2025). At the single-agent level, foundational works have enabled agents to reason and act by interleaving thought and action (Yao et al., 2023b), to enhance complex problem-solving through structured exploration of thoughts (Yao et al., 2023a), and to interact effectively with external tools and APIs (Wu et al., 2024). At the multi-agent level, frameworks such as CAMEL (Li et al., 2023), AutoGen (Wu et al., 2023), and MetaGPT (Hong et al., 2024) have facilitated sophisticated collaboration on complex tasks like software development, demonstrating strong performance across diverse domains. Despite these advances, existing systems remain constrained by a reliance on handcrafted prompts and rigid communication topologies, which limit adaptability as task complexity scales. This has spurred a shift toward automated agentic systems capable of optimizing their own architectures and behaviors.

**Agentic Systems Automation**    Early automation efforts in agentic systems primarily focused on prompt optimization (Pryzant et al., 2023; Ramnath et al., 2025; Li et al., 2025), with approaches such as LLMs-as-optimizers (Yang et al., 2024), self-referential evolution (Fernando et al., 2023), textual gradients (Yuksekgonul et al., 2024), and self-supervised optimization (Xiang et al., 2025). More recent research has expanded beyond prompt-level tuning toward optimizing the architectures and execution flows of entire agentic systems. For example, Liu et al. (2024) explores dynamic communication structures for adaptive collaboration, while Zhuge et al. (2024a) models agents as computational graphs to refine both prompts and inter-agent orchestration. Shang et al. (2024) proposes a novel modular design automatically searching for high-performance agent structures. Zhou et al. (2024) investigates agents capable of self-optimization using symbolic optimizers. Hu et al. (2025) introduces a meta agent that automatically discovers novel, high-performing, and generalizable agentic system designs. Yin et al. (2025) introduces a self-referential framework that enables agents to recursively improve themselves. Zhang et al. (2025b) employs LLMs as optimizers with a Monte Carlo Tree Search (MCTS) variant to discover effective workflows. Zhang et al. (2025a) automatically evolve agentic supernet systems leading to query-specific workflows. Su et al. (2025) leverages debate and reflexion to collaboratively refine workflows while reducing search redundancy. Zheng et al. (2025) introduces safety-constrained evolutionary programming in a declarative graph space, ensuring structural validity and robustness. While these efforts mark significant progress, most existing approaches still focus on end-to-end or global architectural optimization, often leading to inefficient search and a lack of fine-grained diagnostic feedback, which limits both scalability and interpretability as task complexity grows.

**LLM as a Judge**    The LLM-as-a-judge paradigm leverages large language models to automate the evaluation of generated content, addressing the scalability limitations of human assessment (Gu et al., 2025). This approach has been widely adopted for assessing complex outputs based on predefined criteria like correctness, relevance, or rule compliance (Li et al., 2024). However, the effectiveness of the LLM-as-a-Judge framework may be limited by inherent biases in LLMs (Wang et al., 2023a). To mitigate these issues, various methods have been proposed. Liu et al. (2025b) propose a ranking-based alignment method that significantly improves the judging performance of LLMs. In addition, (Zhuge et al., 2024b) proposed the framework to use agentic systems to evaluate agentic systems. In a related application, (Zhang et al., 2025c) attempts to automate the failure attribution for LLM multi-agent systems. Their findings reveal that providing stronger ground-truth signals can substantially improve attribution quality, and aggregated analysis across multiple failures can uncover reliable error patterns.

## 3 METHODOLOGY

### 3.1 PROBLEM FORMULATION

Our framework models an agentic workflow by hierarchically composing basic agentic actions (Operators) into structured logical units (Blocks) as follows.

A configured operator $O(D)$ is the basic unit of agentic action, where $O$ represents a categorical label for its core function like `generate` or `self_refine` (details in Appendix A), and $D$ is the operator configuration, which includes the LLM backbone, prompt template, and other hyperparameters (Zhang et al., 2025b). Building upon operators, a logic block $(B, C)$ is a higher-level structural unit that orchestrates one or more configured operators, where $B \in \mathcal{B}$ is the logic block type, dictating how the operators are orchestrated. The set of available types $\mathcal{B}$ includes three fundamental forms of logic as shown in Fig. 2 (details in Appendix B):

- **SequenceLogic (`seq`)**: A sequential execution block where operators are executed one after another. Each operator consumes the output of its predecessor, ensuring a linear flow of intermediate results until the final operator produces the block output.

- **LoopLogic (`for`)**: An iterative block that repeatedly invokes its internal operators. The iteration continues until the stopping condition is satisfied.

- **ConditionalLogic (`cond`)**: A branching block that first executes a designated condition operator. Based on the evaluation outcome, it then activates one of two operator sequences. Only the operators in the selected branch are executed to generate the block output.

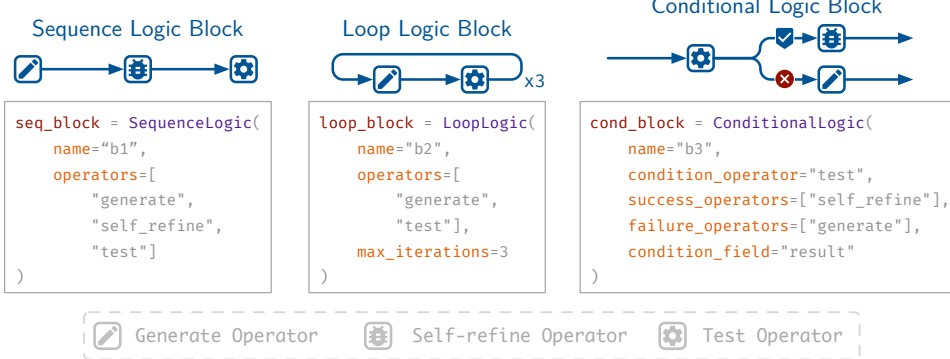

Figure 2: The illustration of logic blocks.

Correspondingly, $C$ is the logic block configuration, which contains the set of configured operators $O(D)$ in the block and block-level parameters (e.g., stopping condition in LoopLogic). Finally, the agentic workflow $W$ is defined as a tuple $W = \left( \{(B_i, C_i)\}_{i=1}^{M}, S \right)$, where $M$ is the total number of logic blocks in the workflow, and $S$ denotes the ordered sequence of logic blocks at the top level while each individual block may internally contain conditional or iterative control. This definition not only preserves the common logic patterns in code-represented workflows ensuring expressive diversity (Hu et al., 2025; Zhang et al., 2025b), but also enhances interpretability, including the explicit semantic characteristics of each logic block and the overall execution trajectory of the workflow, which facilitates subsequent optimization.

Given an input query $q$ from the dataset $\mathcal{D}$ which is available to every block, the execution function $\phi_{\text{exe}}$ processes workflow $W$ by sequentially applying its logic blocks along the execution order $S$. Each block $(B_i, C_i)$ receives the state from the previous block, $a'_{i-1}$, and produces a new state, $a'_i$, formally defined as:

$$a'_i = \phi_{\text{exe}}^{(i)}(a'_{i-1}, q; B_i, C_i), \quad i = 1, 2, \ldots, M, \tag{1}$$

where $\phi_{\text{exe}}^{(i)}$ is the execution function for block $i$ and $a'_0 = \varnothing$. The final workflow output is obtained as $a'_M$, and then scored by the evaluation function $\phi_{\text{eval}}$ against the ground-truth answer $a$ corresponding to $q$. The objective of agentic workflow optimization is to find the optimal workflow $W^*$ that maximizes expected evaluation performance across the dataset:

$$W^* = \underset{W \in \mathcal{W}}{\arg\max} \, \mathbb{E}_{(q,a) \sim \mathcal{D}} \left[ \phi_{\text{eval}}(a'_M, a) \right], \tag{2}$$

where $\mathcal{W}$ denotes the search space of candidate workflows.

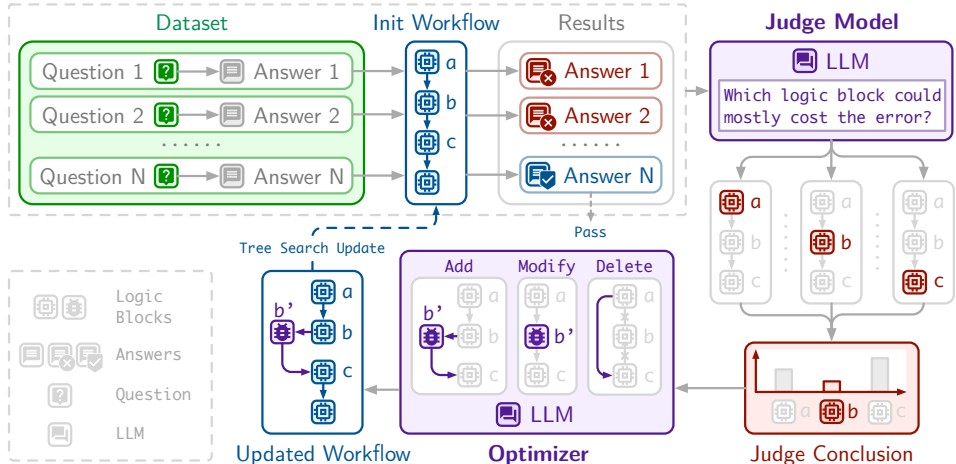

Figure 3: The main pipeline of JUDGEFLOW

## 3.2 JUDGEFLOW

Building on the representation of workflow using logic blocks, JUDGEFLOW incorporates a dedicated Judge module and implements an iterative Evaluation-Judge-Optimization-Update pipeline for the efficient optimization of agentic workflows and continues until a predefined maximum number of optimization rounds are met.

### 3.2.1 EVALUATION-JUDGE

The combined Evaluation-Judge stage, detailed in Algorithm 1, processes each input query from the dataset. If the workflow $W$ fails on a given query, the stage identifies and logs specific problematic block within $W$. This provides targeted diagnostic signals for subsequent workflow optimization, enabling a more efficient and focused approach on refining these identified weak logic to improve overall optimization efficiency.

Specifically, for each input query $q$ (with a corresponding ground-truth answer $a$), we have $\{a_i'\}_{i=1}^M = \phi_{\text{exe}}(q, W)$, and score $s = \phi_{\text{eval}}(a_M', a)$. The score $s$ is recorded in a list $\mathcal{P}_{\text{scores}}$ for later calculation of $W$'s overall performance. Providing a threshold $\varepsilon$ that indicates successful execution, if $s \geq \varepsilon$, the instance is marked as successful, and the algorithm simply proceeds to the next input.

However, if $s < \varepsilon$, a quadruple $Q = (W, q, a, \{a_i'\}_{i=1}^M)$ is defined to encapsulate the full context of the failure. The Judge proceeds to examine the quadruple, assessing each block's $\{B_i\}_{i=1}^M$ responsibility for the failure and ranking them accordingly. This procedure, guided by specific judging prompts (detailed in Appendix C), yields a rank-based score vector (Liu et al., 2025b) $(r_i)_{i=1}^M$ for the blocks where $r_i = 1$ refers to the block deemed most responsible for the failure and $r_i = M$ denotes the least responsible, each rank from 1 to $M$ is assigned exactly once. These block scores $(r_i)_{i=1}^M$ are appended to $\mathcal{R}_{\text{ranks}}$. The RoundWorst$((r_i)_{i=1}^M, W)$ function then utilizes this score vector to identify $B_{\text{rw}}$, the block deemed most problematic for the current instance (i.e. $B_{\text{rw}} = \{B_i \mid r_i = 1\}$). Subsequently, the instance details $(q, a, \{a_i'\}_{i=1}^M)$ are logged into $\mathcal{L}_{B_{\text{rw}}}$, the dedicated log for $B_{\text{rw}}$, providing targeted few-shot examples for its potential future optimization.

Upon completion of all instances in $\mathcal{D}$, the accumulated diagnostic information is processed. The OverallWorst$(\mathcal{R}_{\text{ranks}}, W)$ function analyzes all block rank-based score vectors in $\mathcal{R}_{\text{ranks}}$ to identify $B_{\text{sel}}$, the block deemed the most consistently problematic over the whole dataset. In practice, we aggregate rank vectors across all failing instances in $\mathcal{R}_{\text{ranks}}$ by summing the scores $r_k$ assigned to each block $B_k$, and then selects the block achieving the minimum sum (i.e. $B_{\text{sel}} = \arg\min_{B_k \in W} \sum_{t=1}^T r_k^{(t)}$, where $T$ is the number of the failure executions). Concurrently, the overall performance $P_W$ of $W$ on $\mathcal{D}$ is computed by CalPerformance$(\mathcal{P}_{\text{scores}})$. Finally, this stage returns $P_W$, $B_{\text{sel}}$, and $\mathcal{L}_{B_{\text{sel}}}$, providing actionable insights for subsequent optimization.

---

**Algorithm 1** Evaluation-Judge

---

1: **Input:** Workflow $W$, Dataset $\mathcal{D}$, executor $\phi_{\text{exe}}$, evaluator $\phi_{\text{eval}}$, Judge, threshold $\varepsilon$
2: **Output:** Performance $P_W$, Selected Block $B_{\text{sel}}$ and the corresponding Log $\mathcal{L}_{B_{\text{sel}}}$
3: For $k \leftarrow 1$ to $M$: Initialize $\mathcal{L}_{B_k} \leftarrow \emptyset$
4: $\mathcal{R}_{\text{ranks}} \leftarrow \emptyset, \mathcal{P}_{\text{scores}} \leftarrow \emptyset$
5: **for** each $(q, a) \in \mathcal{D}$ **do**
6:     $\{a'_i\}_{i=1}^M \leftarrow \phi_{\text{exe}}(q, W)$
7:     $s \leftarrow \phi_{\text{eval}}(a'_M, a)$
8:     $\mathcal{P}_{\text{scores}} \leftarrow \text{APPEND}(\mathcal{P}_{\text{scores}}, s)$
9:     **if** $s \geq \varepsilon$ **then**
10:         **continue**             ▷ If success, no judging needed, skipping to next sample
11:     **else**
12:         $(r_i)_{i=1}^M \leftarrow \text{Judge}(W, q, a, \{a'_i\}_{i=1}^M)$      ▷ Call Judge to rank blocks by responsibility
13:         $\mathcal{R}_{\text{ranks}} \leftarrow \text{APPEND}(\mathcal{R}_{\text{ranks}}, (r_i)_{i=1}^M)$          ▷ Append block-wise judge ranking
14:         $B_{\text{rw}} \leftarrow \text{RoundWorst}((r_i)_{i=1}^M, W)$       ▷ Get the most problematic block in this round
15:         $\mathcal{L}_{B_{\text{rw}}} \leftarrow \text{APPEND}(\mathcal{L}_{B_{\text{rw}}}, (q, a, \{a'_i\}_{i=1}^M))$ ▷ Append failure execution context to that block's log
16:     **end if**
17: **end for**
18: $B_{\text{sel}} \leftarrow \text{OverallWorst}(\mathcal{R}_{\text{ranks}}, W)$      ▷ Aggregate across failures to pick the globally weakest block
19: $P_W \leftarrow \text{CalPerformance}(\mathcal{P}_{\text{scores}})$          ▷ Compute overall performance on the dataset
20: **return** $P_W, B_{\text{sel}}, \mathcal{L}_{B_{\text{sel}}}$

---

### 3.2.2 OPTIMIZATION-UPDATE

In the subsequent Optimization-Update stage, the LLM-based optimizer utilizes the insights from the previous stage and refines $W$ to produce an improved version $W'$ guided by specific optimization prompts (detailed in Appendix D), which can be formally expressed as

$$W' \leftarrow \text{Optimizer}(W, B_{\text{sel}}, A, \text{sample}(\mathcal{L}_{B_{\text{sel}}})) \tag{3}$$

where $\text{sample}(\mathcal{L}_{B_{\text{sel}}})$ refers to few-shot samples drawn from the logs $\mathcal{L}_{B_{\text{sel}}}$ and $A \in \mathcal{A}$, where $\mathcal{A}$ is a predefined set of available modification actions as follows:

- **Add Block** : Introduce a new block $B_{\text{new}}$ with configuration $C_{\text{new}}$, and connect it directly with the low-performing block $B_{\text{sel}}$;

- **Remove Block**: Remove the low-performing block $B_{\text{sel}}$ together with all of its incident edges while reconnecting its predecessor and successor to preserve sequential flow;

- **Modify Block**: Reconfigure the existing $B_{\text{sel}}$ by updating its configuration $C_{\text{sel}} \mapsto C'_{\text{sel}}$.

In practice, the LLM-based optimizer selects $A$ adaptively based on the diagnostic signals in $\mathcal{L}_{B_{\text{sel}}}$. Following Zhang et al. (2025b), the refined workflow $W'$ is first evaluated to obtain its performance score $P_{W'}$. The pair $(W', P_{W'})$ is then added to the candidate pool $\mathcal{W}_{\text{pool}}$, which retains at most $K$ workflows by keeping the top-$K$ highest-scoring entries:

$$\mathcal{W}_{\text{pool}} \leftarrow \text{Top-}K\big(\mathcal{W}_{\text{pool}} \cup \{(W', P_{W'})\}\big). \tag{4}$$

At the beginning of the next iteration, the optimizer selects a starting workflow $W_{\text{start}}$ from $\mathcal{W}_{\text{pool}}$ using a softmax distribution over scores with temperature $\tau$:

$$W_{\text{start}} \sim \mathcal{W}_{\text{pool}}, \quad \Pr(W_i) = \frac{\exp\left(\frac{s_i - \max_j s_j}{\tau}\right)}{\sum_{k=1}^{|\mathcal{W}_{\text{pool}}|} \exp\left(\frac{s_k - \max_j s_j}{\tau}\right)}, \tag{5}$$

where $s_i$ is the evaluation score of workflow $W_i$.

## 4 EXPERIMENTS

### 4.1 EXPERIMENTAL SETUPS

**Benchmarks and Datasets.** We evaluate our method on widely used public benchmarks, covering math reasoning tasks (GSM8K (Cobbe et al., 2021), MATH (Hendrycks et al., 2021)) and code generation tasks (MBPP (Austin et al., 2021), HumanEval (Chen et al., 2021)).

Following previous studies (Zhang et al., 2025b;a), each dataset is divided into training and test sets with a ratio of 1:4. We report the solve rate (%) on GSM8K and MATH, pass@1 on MBPP and HumanEval as evaluation metrics.

**Baselines.** We compare our JUDGEFLOW with a series of baselines, including (1) Single-agent System: Standard prompting (IO), Chain-of-Thought prompting (CoT) (Wei et al., 2023), and Self-Consistency (Wang et al., 2023b); (2) Hand-crafted Multi-agent System: MultiPersona (Wang et al., 2024c), SELF-REFINE (Madaan et al., 2023), LLM-Debate (Du et al., 2023), LLM-Blender (Jiang et al., 2023), and DyLAN (Liu et al., 2024); (3) Autonomous Multi-agent System: GPTSwarm (Zhuge et al., 2024a), ADAS (Hu et al., 2025), AFlow (Zhang et al., 2025b), MaAS (Zhang et al., 2025a), and MermaidFlow (Zheng et al., 2025).

**Implementation Details.** We use the closed-source LLM `gpt-4o-mini-0718` (OpenAI, 2024b) as both the optimization LLM and execution LLM following the previous works Zhang et al. (2025a) and Zheng et al. (2025). For a fair comparison, we use the same model as Judge LLM. All the models are accessed via API with temperature = 0. The number of iteration rounds is set to 20 consistent with Zhang et al. (2025b) and Zheng et al. (2025). When optimizing, we set $M \leq 3$, $\varepsilon = 1$, and $K = 3$.

### 4.2 EXPERIMENTAL RESULTS

Table 1: Performance comparison with baselines on **GSM8K**, **MATH**, **MBPP**, and **HumanEval**. The results are averaged over three independent runs.

| Method | GSM8K | MATH | MBPP | HumanEval | Avg. |
|---|---|---|---|---|---|
| *Single-agent System* | | | | | |
| IO | 87.8 | 48.6 | 73.9 | 87.0 | 74.3 |
| CoT (Wei et al., 2023) | 87.0 | 48.8 | 74.2 | 88.6 | 74.7 |
| CoT SC (Wang et al., 2023b) | 86.9 | 50.4 | 73.3 | 91.6 | 75.6 |
| *Hand-crafted Multi-agent System* | | | | | |
| SELF-REFINE (Madaan et al., 2023) | 85.5 | 46.1 | 71.8 | 87.8 | 72.8 |
| LLM-Debate (Du et al., 2023) | 89.5 | 48.6 | 70.3 | 88.8 | 74.3 |
| LLM-Blender (Jiang et al., 2023) | 88.4 | 46.9 | 77.1 | 88.7 | 75.3 |
| DyLAN (Liu et al., 2024) | 90.0 | 48.5 | 77.3 | 90.4 | 76.6 |
| *Autonomous Multi-agent System* | | | | | |
| GPTSwarm (Zhuge et al., 2024a) | 89.1 | 47.9 | 77.4 | 89.3 | 75.9 |
| ADAS (Hu et al., 2025) | 88.4 | 43.2 | 77.1 | 84.2 | 73.2 |
| AFlow (Zhang et al., 2025b) | 90.1 | 52.8 | 81.7 | 90.1 | 78.7 |
| MaAS (Zhang et al., 2025a) | 91.5 | 52.2 | 82.2 | 91.6 | 79.4 |
| MermaidFlow (Zheng et al., 2025) | 92.4 | 55.4 | 82.3 | 92.9 | 80.8 |
| **JUDGEFLOW (Ours)** | **93.0** | **58.5** | **83.8** | **93.4** | **82.2** |

**Main Results.** As shown in Table 1, JUDGEFLOW achieves superior performance compared to several strong baselines, including both hand-crafted and autonomous multi-agent systems consistently across all the tasks[1]. Notably, for some challenging benchmarks such as MATH and MBPP, JUDGEFLOW outperforms the strongest prior baseline MermaidFlow by +3.1(5.6%) and +1.5(1.8%),

---

[1]Some baseline results are referred to Zhang et al. (2025b) and Zheng et al. (2025).

respectively. At the same time, for relatively simpler benchmarks such as GSM8K and HumanEval, JUDGEFLOW still achieves consistent gains of +0.6 and +0.5. Taken together, JUDGEFLOW achieves the average score of 82.2, representing a +1.4(1.7%) increase. The results highlight the effectiveness of our Judge-guided block-level optimization across both reasoning and code generation tasks.

**Main Results.** While several works in this area (Zhang et al., 2025b; Ma et al., 2025; Zheng et al., 2025) compare on standard benchmarks to ensure fair comparison, we extended our evaluation to significantly more challenging AIME benchmark (Ye et al., 2025) to demonstrate the performance of JUDGEFLOW on complex reasoning tasks.

To balance performance and cost for this more challenging benchmark, we employed `gpt-4.1-mini` (OpenAI, 2025) as the LLM backbone for the Judge, Optimization, and Execution modules. Following previous works and our standard setting, the temperature was set to 0. We sampled validation data from the AIME 2024 dataset during the optimization phase and evaluated the final performance on the AIME 2025 dataset. We evaluated the workflows across 5 runs. As shown in Fig. 4, JudgeFlow achieves an average accuracy of 44.67%, improving AFlow's absolute performance by 2.67 percentage points.

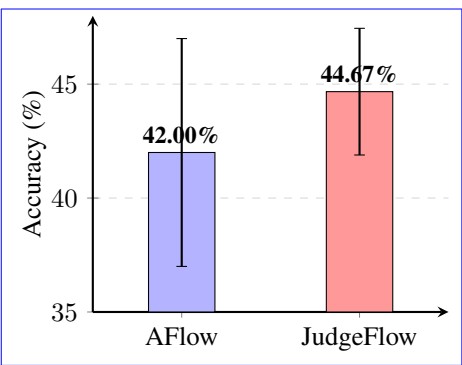

Figure 4: Performance on AIME 2025.

### 4.3 ANALYSIS

We take the MBPP dataset as an illustrative example to analyze JUDGEFLOW.

**Best-Performing Workflow.** Fig. 5a is the best-performing workflow found by JUDGEFLOW on MBPP dataset. The workflow is composed of three logic blocks. First, a `seq` block b1 applies a `generate` operator to produce an initial candidate function. Second, a `for` block b2 repeatedly invokes the `test` operator until the stopping condition is satisfied. Finally, a `cond` block b3 runs the `test` operator to check correctness: if the candidate doesn't pass, it routes the solution to a `self_refine` operator for further improvement.

**Learning Curves.** Fig. 5b compares the training curve and testing curve of the highest performance found between JUDGEFLOW and AFlow. JUDGEFLOW exhibits clear performance gains within the first five optimization iterations, with both the training and testing curves showing rapid improvements. Beyond this early stage, JUDGEFLOW continues to achieve gains, ultimately converging to higher accuracy. In contrast, AFlow remains stagnant across most iterations and only shows noticeable improvements in the later stage, and its final training and testing performance remain consistently lower than those of JUDGEFLOW.

**Impact of LLMs.** According to Table 2, we keep `gpt-4o-mini-0718` fixed as the executor LLM, while varying the optimization and Judge models. Particularly, we consider `gpt-4o` (OpenAI, 2024a) and `Gemini-2.5-flash` (Google-Cloud, 2025) as alternatives for these roles and report the resulting performance. The experiment confirms that increasing the capacity of optimization and Judge models consistently improves performance. While all models yield competitive results, GPT-4o attains the best score 84.5.

Table 2: Testing performance using different LLMs on MBPP dataset.

| Models | Score |
|---|---|
| GPT-4o-mini | 83.8 |
| GPT-4o | 84.5 |
| Gemini-2.5-flash | 84.4 |

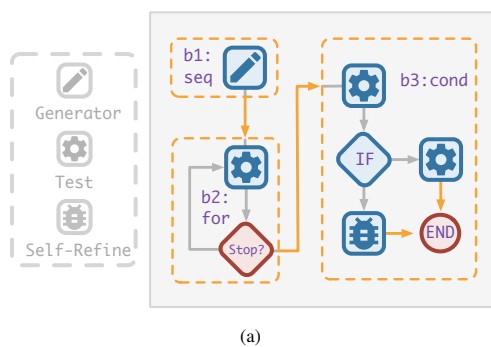 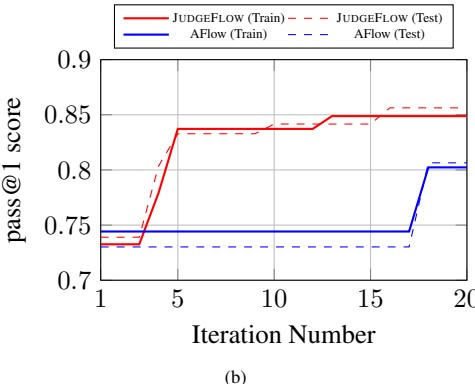

(a)                                          (b)

Figure 5: Fig. 5a The optimal workflow found by JUDGEFLOW on MBPP dataset; Fig. 5b The training and testing curve between JUDGEFLOW and AFlow on MBPP dataset.

**Optimization Efficiency.** We perform an ablation study on different key components of JUDGEFLOW. As shown in Table 3, removing the logic block abstraction or the judge module leads to consistent performance drops, confirming the importance of both design choices.

Table 3: Ablation results on MBPP.

| Method | Score |
|---|---|
| JUDGEFLOW | 83.8 |
| - Logic Block | 81.8 |
| - Judge | 80.6 |

**Cross-benchmark Generalization.** To demonstrate the generalization of JUDGEFLOW, we evaluate the cross-benchmark transferability, including the Math transfer and Code transfer. We optimize the workflow based on the MATH(MBPP) dataset and zero-shot evaluation on GSM8K(HumanEval) dataset. We report the solve rate (%) on GSM8K and pass@1 on HumanEval as evaluation metrics in Table 4, showing JUDGEFLOW yields state-of-the-art results.

Table 4: Cross-Benchmark Transfer Performance

| Optimization → Evaluation | MATH → GSM8K | MBPP → HumanEval |
|---|---|---|
| AFlow | 91.95 | 90.84 |
| JUDGEFLOW | **92.89** | **93.89** |

**Judge** To ensure robustness against noisy outputs, JUDGEFLOW employs a statistical filtering mechanism. Instead of a single potentially noisy judge output, JUDGEFLOW collects traces from all failed instances and aggregates responsibility scores using the `OverallWorst` mechanism. We follow the recent finding in (Zhang et al., 2025c); while individual LLM-based failure attribution might contain noise, the aggregated distribution of responsible logic blocks is more consistent with the true causes of failure.

Regarding computational cost, although JUDGEFLOW introduces an additional Judge module for LLM calls, our analysis reveals that the dominant cost in agentic workflow optimization lies in the Evaluation phase rather than the Judge module in our proposed methods. For example, we monitor the cost for a single optimization round on the GSM8K dataset, and the Evaluation costs $0.45 while the Judge costs $0.01, the ratio of Judge/Evaluation is about 2%.

**Logic Block** The Logic Block serves as a critical abstraction designed to enable the Judge mechanism. At the operator level, operators often interact through complex control flows (e.g., the `if-else` structure in block b3 of Fig. 5a), making it difficult to attribute blame to a single operator because execution paths vary dynamically per query (e.g., an operator in an unexecuted else branch). By introducing the Logic Blocks, we encapsulate these dynamic control flows into a stable semantic unit. This abstraction mitigates the ambiguity caused by varying execution paths, thereby enabling the Judge to perform stable and effective attribution.

## 4.4 CASE STUDY

To illustrate how JUDGEFLOW works in practice, we present a case study of workflow optimization on the GSM8K dataset. This example demonstrates how JUDGEFLOW automatically identifies and rectifies a suboptimal workflow through the pipeline, as shown in Fig. 6. The initial workflow consists of two logic blocks: b1, a seq block consisting of one multi_generate_ensemble operator designed to generate and ensemble multiple candidate solutions (with num_solutions set to 3), and b2, a seq block consisting of one programmer operator, which takes the output from the previous block and generates the final answer using programming. When processing a batch of GSM8K instances, this workflow failed multiple times, triggering the Evaluation-Judge stage. The Judge module analyzed execution traces of these failures and assigned rank-based responsibility scores to each block. For example, in one failed run, it output {"b2": 1, "b1": 2}, attributing the primary blame to b2, while in another it output {"b1": 1, "b2": 2}, assigning higher responsibility to b1. By aggregating these rank-based scores across failures, the system identified b1 as the OverallWorst block, indicating that low-quality initial solutions from b1 were the main bottleneck, making it difficult for the workflow to generate correct final answers.

In the Optimization-Update stage, the LLM-based Optimizer received this diagnostic signal and selected the *Add Block* action. It introduced a new logic block, b3, of type seq, with operator self_refine, which iteratively improves candidate solutions. This block was inserted between b1 and b2, producing the new workflow ["b1", "b3", "b2"]. The updated workflow first generates multiple ideas with b1, then refines them with b3, and finally produces the polished answer through b2. This case study highlights the strength of JUDGEFLOW: instead of relying solely on end-to-end success signals, it leverages block-level diagnostics from the Judge to perform precise error attribution, enabling workflow modifications that directly address weaknesses. As a result, JUDGEFLOW avoids blind search, achieves more efficient optimization, and substantially improves performance.

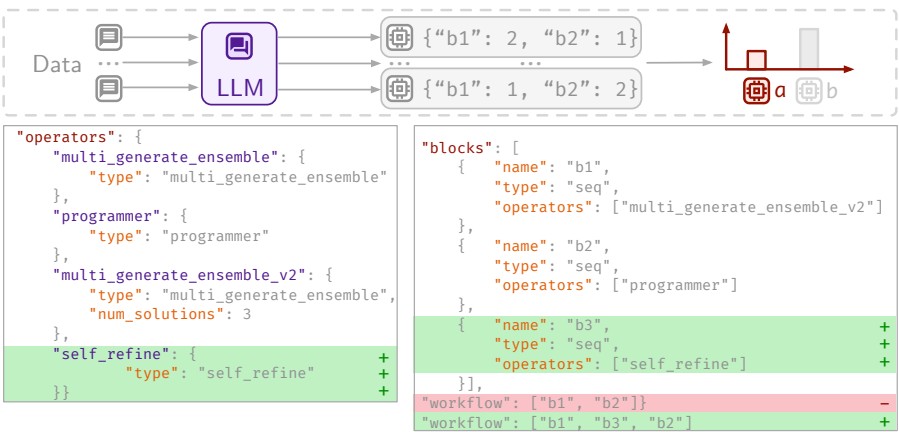

Figure 6: The illustration of the case study in the GSM8K dataset.

## 5 CONCLUSION

In this paper, we presented a novel Evaluation-Judge-Optimization-Update pipeline named JUDGE-FLOW for automating the optimization of agentic workflows. By introducing reusable logic blocks as higher-level structural abstractions, JUDGEFLOW achieves a balance between the expressive flexibility of code-based workflows and the tractability of optimization. On top of this representation, the Judge module provides block-level diagnostic signals by analyzing execution traces and assigning responsibility to problematic block, enabling more interpretable and fine-grained optimization. Through extensive experiments on mathematical reasoning and code generation benchmarks, we demonstrate that JUDGEFLOW consistently outperforms strong baselines. While achieving success in optimizing agentic workflows, LLM-as-a-Judge can be biased and may provide misleading responsibility scores. Future work may include exploring more robust Judge for agentic systems optimization, such as statistical signals or other validation methods.

REPRODUCIBILITY STATEMENT

To enable the independent reproducibility of our results, we provide complete access to our implementation. The source code is available at https://anonymous.4open.science/r/JudgeFlow. Detailed descriptions of the framework, models, and experimental settings are provided in the main paper and its appendix.

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

## A  OPERATORS

Following Zhang et al. (2025b), Zhang et al. (2025a) and Zheng et al. (2025), we adopt the following set of operators:

1. `generate`, a generation operator that produces candidate solutions based on the problem description and optional previous results.

2. `test`, a testing operator that executes generated solutions against test cases and provides feedback for refinement.

3. `self_refine`, a refinement operator that improves a given solution through self-refinement.

4. `multi_generate_ensemble`, an ensemble operator that generates multiple solutions and combine them to the best one via self-consistency.

5. `programmer`, a synthesis-and-execution operator that generates Python code for solving math problems, runs it in a restricted environment, and iteratively repairs errors.

## B  LOGIC BLOCKS

We implement three common logic types in code-represented workflows: **SequenceLogic (seq)**, **LoopLogic (for)**, and **ConditionalLogic (cond)**, whose descriptions and interfaces are listed below.

**Logic Blocks**

```
{
    "SequenceLogic": {
        "type": "seq",
        "description": "Execute operators strictly in order. Required
            fields: name (string), type (must be 'seq'), operators (
            array of operator aliases). No optional fields. Use this
            for linear processing flows where you need sequential
            execution of operators.",
        "structure": {
            "name": "block_name",
            "type": "seq",
            "operators": ["operator"]
        },
        "input_flow": "block_input -> op1 -> op2 -> ... ->
            block_output"
    },
    "LoopLogic": {
        "type": "for",
        "description": "Iteratively execute a sequence of operators
            until the optional asynchronous condition returns False or
             the max iteration limit is reached. Required fields: name
             (string), type (must be 'for'), operators (array of
            operator aliases). Optional fields: max_iterations (
            integer, default 3), condition (object with 'field' and '
            equals' properties, or null for no condition). Use this
            for retry mechanisms and iterative refinement.",
        "structure": {
            "name": "block_name",
            "type": "for",
            "operators": ["operator"],
            "max_iterations": num_iterations,
            "condition": {
                "field": "field_name",
                "equals":
            }
        },
        "input_flow": "block_input -> repeat [op1 -> op2 -> ...] until
            stop -> block_output"
```

```
    },
    "ConditionalLogic": {
        "type": "cond",
        "description": "Run a dedicated condition operator first, then
            choose the success or failure branch based on the field
            specified by 'condition_field'. The chosen branch runs
            sequentially with the same data-passing semantics as
            SequenceLogic. Required fields: name (string), type (must
            be 'cond'), condition_operator (string, operator alias to
            evaluate condition), success_operators (array of operator
            aliases for success path), failure_operators (array of
            operator aliases for failure path). Optional fields:
            condition_field (string, field name to check for condition
             result, default 'result'). The condition operator
            evaluates criteria and sets a result field, which
            determines whether to execute success_operators or
            failure_operators. Use this for branching logic and
            conditional processing. ",
        "structure": {
            "name": "block_name",
            "type": "cond",
            "condition_operator": "condition_operator",
            "success_operators": ["success_op"],
            "failure_operators": ["failure_op"],
            "condition_field": "field_name"
        },
        "input_flow": "block_input -> condition operator -> select
            branch -> branch sequence -> block_output"
    }
}
```

## C    JUDGE PROMPT

> **Judge Prompt**
>
> You are a workflow failure analyst. Given execution evidence from a block-based AI
> ↪  workflow that produced an incorrect answer, determine which logic block is
> ↪  causally responsible for the failure.
>
> # Knowledge Base
> ## Logic block types
> {logic_block_descriptions_text}
>
> ## Operator types
> {operator_descriptions_text}
>
> # Responsibility Principles:
> - Consider blocks that **actually make mistakes** over blocks that only perform
> ↪  **redundant** work.
> - Our goal is to identify the weakest block in this workflow, so that in later
> ↪  optimization we can focus on improving this weakest block.
> - You will be given: the **problem**, the **correct answer**, the **incorrect
> ↪  answer**, the **workflow execution trace**, and **each block's inputs/outputs**
> ↪  in a **sequential** pipeline. Ground your judgment in this evidence:
>     - For each block *i*, compare its **output vs. input**, and **output vs. the
>     ↪  correct answer** to locate where the **first critical deviation** was
>     ↪  introduced, how later blocks **propagated/amplified** it, and whether any
>     ↪  block **had enough information to correct** it but failed to do so.
>     - Do **not** overweight temporal order:

```
     - **Earlier** blocks bear more responsibility for **introducing** the critical
     ↪  error.
     - **Later** blocks bear responsibility for **failing to correct** earlier errors
     ↪  **given the available context**.
   - If two blocks seem equally responsible, apply **counterfactual** reasoning: *If
   ↪  this block were correct, would the final answer be correct?*
   - You may form a brief **internal** natural-language reason (e.g., "this block
   ↪  generated incorrect code") to aid the decision, but the **output must be JSON
   ↪  only**.
```

## D OPTIMIZATION PROMPT

**System Prompt**

```
You are an expert workflow optimization assistant specializing in Logic Block-based
↪  AI workflows for the {{dataset}} dataset.

IMPORTANT: Focus exclusively on optimizing the low-performing logic block to improve
↪  code generation quality and overall workflow performance.
IMPORTANT: You have exactly one optimization attempt. Reason carefully and aim to
↪  improve performance across the entire dataset.

# Task Overview

You will be provided with:
1. Error examples showing: problem, correct answer, workflow's wrong answer, and the
↪  low-performing block's output
2. Current workflow definition
3. Performance analysis results

Your objective: Optimize the identified low-performing logic block using the error
↪  examples as guidance while avoiding overfitting.

# Logic Block Types and Detailed Semantics
{logic_blocks_section}

# Available Operators
{operators_section}

# Critical Instructions for Operator Usage

**INSTRUCTION Field is Crucial**:
- The `instruction` field is extremely important for operator performance and
↪  directly impacts final output quality
- Instructions should clearly guide the operator on how to process input and produce
↪  expected output
- For code generation tasks, instructions need to include specific programming
↪  requirements, output format, and quality standards
- For mathematical reasoning tasks, instructions need to include specific
↪  problem-solving approaches, step-by-step reasoning requirements, and output
↪  format standards

# Optimization Strategies

Choose exactly one strategy:

## 1. Add Block Strategy
- Create a completely new logic block with its own name (e.g., "b2", "b3")
- Insert the new block immediately before or after the low-performing block
- Select appropriate block type (seq/for/cond) that complements the low-performing
↪  block
```

```
- Populate all required parameters (instructions, iteration limits, condition
↪  fields, etc.)
- Run internal counterfactual reasoning but do not output explorations

Example: `"workflow": ["b1", "b2"] ("b2" performs worst) → "workflow": ["b1", "b2",
↪  "b3"]`

## 2. Remove Block Strategy
- Completely delete the low-performing block when it adds noise or harms outcomes
- Internally evaluate workflow behavior without that block
- Update workflow sequence and remove unused operators

Example: `"workflow": ["b1", "b2"] ("b1" performs worst) → "workflow": ["b2"]`

## 3. Modify Block Strategy
- Rework the existing low-performing block without introducing new blocks
- Examine block's logic type, operator choices, and parameterization
- Update operators, ordering, and configuration for stronger reasoning
- Focus solely on refining the current block

# Critical Constraints

CRITICAL: Maximum 3 blocks per workflow - DO NOT EXCEED this limit
CRITICAL: Create NEW BLOCK with different name when adding
IMPORTANT: Focus on the low-performing block identified in the analysis
IMPORTANT: Maintain compatibility with other blocks in the workflow
IMPORTANT: Each block should have a clear, distinct purpose

# Prohibited Actions

- NEVER reproduce workflow configurations matching provided history
- MUST NOT repeat, reuse, or recycle any optimization from Previous Optimization
↪  Analysis
- All workflows in previous optimization analysis are explicitly banned
- Run internal "novelty check" to confirm at least two structural differences from
↪  banned workflows

# Output Requirements

- Apply exactly one modification strategy (Add/Remove/Modify)
- Focus only on the identified low-performing logic block
- Output clean JSON without comments or explanations
- Ensure JSON is fully parseable and syntactically correct
- Avoid overfitting to provided error examples
```

**User Prompt**

```
## Dataset
<dataset>{dataset}</dataset>

## Current Workflow Performance
Current workflow score: <score>{score}</score>

Low-performing logic block identified:
<low_performing_blocks>{low_performing_blocks}</low_performing_blocks>

## Current Workflow Definition
```json
<previous_code>{previous_code}</previous_code>
```
```

```
## Error Analysis
Error examples show:
- **Problem**: Original code generation task/question
- **Correct Answer**: Expected output
- **Workflow Wrong Answer**: Current workflow output
- **Low-performing Block Output**: Problematic block's specific output

## Previous Optimization History
STRICTLY PROHIBITED: Do not repeat or reuse any optimization results below.
<reflection_result>{reflection_result}</reflection_result>

IMPORTANT: All workflows above and current definition are disallowed baselines.

# Optimization Task

Analyze the low-performing logic block and improve its output quality.

## Core Optimization Objective
**Your optimization purpose is to modify the weakest block:**
- Deeply analyze why this weak block led to the final incorrect answer
- Understand the block's role and impact within the entire workflow
- Identify the specific failure patterns and root causes of this block
- Your chosen action (Add/Modify/Remove) should be aimed at solving the current
↪   problems

## Key Focus Areas
- Low-performing block is your primary optimization target
- Use error cases to understand failure patterns
- Improve block's reasoning or processing capability
- Evaluate block type appropriateness (seq/for/cond)
- Assess operator suitability and configuration
- **Pay special attention to the quality and detail of instruction fields**

## Strategy Guidelines
Current workflow has
↪   <workflow_block_count>{workflow_block_count}</workflow_block_count> block(s).

## Error Examples
Use these to understand failures, but avoid overfitting:
<error_cases_section>{error_cases_section}</error_cases_section>

# Final Instruction
Generate the optimized JSON workflow definition:
```

# E    USE OF LARGE LANGUAGE MODELS

LLMs played a crucial role in our paper, as we utilized them for workflow optimization. Outside of this usage, we have used LLMs as writing assistants for improving clarity, style, and grammar and as coding assistants. Notably, the core research contributions—among which the design of the framework and validation of results—were conceived and verified exclusively by the authors. All outputs from LLMs were critically assessed, refined, and integrated to ensure correctness and adherence to academic standards.

