# OpenReview forum: "JudgeFlow: Agentic Workflow Optimization via Block Judge"
_ICLR.cc/2026/Conference — Submitted to ICLR 2026_

### Official Review · Reviewer_hjhQ · 2025-10-31

**Soundness:** 2
**Presentation:** 3
**Contribution:** 2
**Rating:** 6
**Confidence:** 4

**Summary:**

This paper introduces JudgeFlow, a framework for enhancing the optimization effect of agentic workflow by using llm-as-a-judge for attribution. The core contributions include (1) introducing reusable logic blocks as higher-level structural abstractions and attribution unit, (2) designing a Judge module that analyzes execution traces to assign rank-based responsibility scores to attribution unit, and (3) leveraging the attribution to guide targeted workflow optimization. With experiments on multiple benchmarks, JudgeFlow demonstrates improvements over existing methods including hand-crafted and autonomous multi-agent systems.

**Strengths:**

- **Sufficient Motivation**: The paper identifies an important gap in existing methods that rely only on end-to-end evaluation signals. The key insight is that failure attribution is critical for effective optimization. The Judge module provides block-level diagnostic signals by analyzing execution traces, enabling enables targeted optimization instead of blind exploration.
- **Comprehensive Experiments**: The paper evaluates on multiple benchmarks, including mathematical reasoning and code generation. Results show consistent improvements across all tasks. The ablation studies validate the importance of both logic blocks and the judge module.
- **Clear Presentation**: The paper is well-written with clear motivation and good use of figures. The case study effectively illustrates how the approach works in practice.

**Weaknesses:**

- **Limited Novelty in Attribution Strategy**: The Judge module is a straightforward application of llm-as-a-judge, which lacks technical depth and novelty. The rank-based scoring mechanism is simple and may not capture complex failure patterns. More sophisticated attribution methods could potentially provide better optimization signals.
- **Insufficient Analysis of Judge Module**: The paper acknowledges that llm-as-a-judge may be biased but provides no empirical validation. There is no ground-truth comparison to verify whether the judge correctly identifies problematic blocks. This is critical for trusting the optimization process.
- **Missing Cost Analysis**: The paper provides no analysis of optimization or execution cost. The Judge module requires additional LLM calls for every failed instance during evaluation. Combined with the optimization loop, the total number of LLM calls and token cost could be substantial. Without cost comparison against baselines, the practical value of the modest performance gains (1.4\% average) is unclear.

**Questions:**

- Could similar judge-based approaches work at the operator level? Is there empirical evidence that blocks are essential for effective judging?
- The paper fixes block abstraction at a specific level (between operators and full workflows). Is there an optimal granularity for attribution?

---

> ### Author Response · Authors · 2025-11-27
>
> Thanks for your encouragement acknowledgement of strong motivation, solid experiment, and clear presentation. We will address your concerns as follows.
>
> **Novelty of Attribution Strategy**
>
> We thank the reviewer for pointing out the insightful feedback. We respectfully clarify that our core contribution lies in the JudgeFlow framework, specifically the abstraction of Logic Blocks and the Judge module for agentic workflow optimization. We adopted the LLM-as-a-judge and ranking-based mechanisms as a foundation of JudgeFlow, and move forward by incorporating these Judge signals for agentic workflow optimization.
>
> **Judge Analysis**
>
> In the context of automated optimization, the "ground truth" of the Judge is best measured by the effectiveness of the resulting optimization. If the Judge were identifying the wrong blocks, the Optimizer would "fix" non-broken parts, leading to stagnant performance. However, our Ablation Study (Table 3) shows that although there is no explicit ground-truth attribution label, the optimization outcome indirectly validates that the judge identifies meaningful failure sources.
>
> At the same time, to enhance the Judge Module, instead of a single potentially noisy judge output, JudgeFlow collects traces from all failed instances and aggregates responsibility scores using the `OverallWorst` mechanism. We follow the recent finding in [1]; while individual LLM-based failure attribution might contain noise, the aggregated distribution of responsible logic blocks is more consistent with the ground truth.
>
> **Cost Analysis**
>
> While JudgeFlow introduces an additional Judge module for LLM calls, our analysis reveals that the dominant cost in agentic workflow optimization lies in the Evaluation phase rather than the Judge module in our proposed methods. For example, we monitor the token consumption and cost (`gpt-4o-mini`) for a single optimization round on the GSM8K dataset, and the Evaluation costs \\$0.45 and the Judge costs \\$0.01, the ratio of Judge/Evaluation is only 2%. Also, as shown in Figure 5b in the paper, JudgeFlow converges significantly faster compared to the baseline, leading to more efficient optimization.
>
> **Analysis of Logic Block Level Judge**
>
> We believe that Logic Blocks are essential and the introduction of Logic Blocks is an abstraction designed to enable our Judge mechanism. For the judgment of operator-level, operators often interact through complex control flows (e.g., if-else branches), leading to attributing blame to a single operator being difficult because execution paths vary dynamically per query (e.g., an operator in an unexecuted else branch). By introducing the Logic Blocks, we encapsulate these dynamic control flows into a stable semantic unit, where the Judge is able to effectively assign responsibility (e.g., for if-else branches, the Judge will not be confused by different execution paths), providing more stable signals that guide the optimization.
>
> For more details, please kindly refer to our revised manuscript.
>
> ---
>
> [1] Zhang, Shaokun, et al. "Which Agent Causes Task Failures and When? On Automated Failure Attribution of LLM Multi-Agent Systems." Forty-second International Conference on Machine Learning (2025).

---

### Official Review · Reviewer_jeni · 2025-11-01

**Soundness:** 3
**Presentation:** 3
**Contribution:** 3
**Rating:** 4
**Confidence:** 5

**Summary:**

This paper addresses a critical limitation in agentic workflow optimization: the lack of fine-grained diagnostic signals to guide the optimization process. By introducing a Judge module that performs block-level error attribution, JudgeFlow can identify problematic components within workflows and focus modifications accordingly. The experimental results demonstrate superior performance and sample efficiency compared to existing methods like AFlow. Overall, the paper is well-written with clear presentation and makes a solid contribution to automated workflow optimization.

**Strengths:**

S1. Identifies a key problem: The paper correctly identifies that prior workflow generation methods rely heavily on heuristic optimization with LLMs alone, lacking concrete optimization signals to guide the search process effectively.

S2. Improved optimization efficiency: The Judge-guided error attribution mechanism significantly enhances both efficiency and effectiveness of workflow optimization. As shown in Figure 4b, JudgeFlow achieves better performance than baseline AFlow with fewer optimization iterations.

S3. Novel evaluation-judge paradigm: The separation of evaluation and judgment provides actionable diagnostic information, enabling targeted modifications rather than blind search across the entire workflow space.

S4. Clear presentation: The paper is well-written with good organization. The methodology is clearly illustrated (e.g., Figure 3 effectively visualizes the evaluation-judge-optimization-update pipeline), making the contributions easy to follow.

**Weaknesses:**

S1. Computational efficiency: The optimization process requires evaluating all samples in the dataset at each iteration, which can be computationally expensive. A potential improvement would be to gradually reduce the validation set size during optimization to lower costs.

S2. Incremental contribution: While adding the Judge module is useful, the overall novelty appears limited. The paper primarily introduces optimization signals but lacks deeper insights or substantial architectural innovations beyond existing frameworks like AFlow and ADAS. Nevertheless, it remains a solid piece of work.

S3. Outdated benchmarks: The evaluation datasets (GSM8K, MATH, MBPP, HumanEval) are somewhat dated. Including more recent and challenging benchmarks such as LiveCodeBench or AIME would strengthen the paper's contribution and better demonstrate the method's capabilities on harder problems.

**Questions:**

Please refer to the weak points.

---

> ### Author Response · Authors · 2025-11-27
>
> Thank you very much for your careful review and valuable suggestions. We will address your concerns as follows.
>
> **[S1] Computional Efficiency**
>
> We thank you for the insightful feedback and the promising way to further improve efficiency. In our JudgeFlow, while we evaluate all samples, the Judge module only activates for failed instances, and as the performance increases, the number of failures drops naturally. At the same time, we respectfully claim that using the full dataset ensures that the aggregated Judge signal obtained by `OverallWorst` mechanism is statistically significant, while reducing the sample size may potentially increase the variance of the diagnostic signal, leading to the optimizer overfitting to certain samples rather than the whole dataset. For statistical stability, we use the full dataset in our experiments. We acknowledge that implementing some adaptive sampling strategies is a valuable direction, as the reviewer mentioned.
>
> **[S2] Clarification of Novelty**
>
> We sincerely thank you for recognizing our paper as a solid piece of work. Regarding the concern about incremental novelty, we respectfully clarify our insights about the introduction of the Logic Blocks and Judge module, and the whole JudgeFlow framework.
>
> 1. **Many existing code-presented workflow optimization lacks structural interpretability.** While code-presented workflows can offer a vast search space, optimizing these workflows is still challenging due to the lack of structural interpretability. Previous end-to-end methods like AFlow only provide the final "binary" feedback for each evaluation, leading to the absence of directional guidance within the workflow.
> 2. **Operator-level attribution is unstable due to dynamic control flow.** In code-presented workflows, operators often interact through complex control flows (e.g., if-else branches), leading to attributing blame to a single operator being difficult because execution paths vary dynamically per query (e.g., an operator in an unexecuted else branch), potentially leading to noisy and inconsistent diagnostic signals across different data samples.
> 3. **Logic Blocks serve as the stable semantic unit for the Judge.** By introducing the Logic Blocks, we encapsulate these dynamic control flows into a stable semantic unit, where Judge is able to assign more stable attribution without the ambiguity caused by different execution paths.
>
> **[S3] Additional Benchmark**
>
> Following previous works such as AFlow[1] and MermaidFlow[2], we primarily used HumanEval and MBPP as our benchmarks to ensure a fair comparison.  However, we agree with the reviewer that harder benchmarks are important. On AIME, which is substantially more challenging than GSM8K/MATH, JudgeFlow improves the performance over AFlow. The results are as follows:
>
>
> | Method     | Result (Mean ± Std Dev) |
> |------------|--------------------------|
> | AFlow      | 0.4200 ± 0.0460          |
> | JudgeFlow  | 0.4467 ± 0.0278          |
>
>
> In the additional experiments presented in the revised manuscript, we directly utilized the official implementation of AFlow and our implementation of JudgeFlow. To balance performance and cost for this harder benchmark, different from `gpt-4o-mini` used in our original paper, we used `gpt-4_1-mini-2025-04-14` as both Judge, Optimization, and Execution LLM backbone. Following the previous work and our paper, we set the temperature to 0. We sampled the validation data from the AIME 2024 dataset during optimization and evaluated the performance on the AIME 2025 dataset. Given the limited sample size of AIME, we established a validation-to-test ratio of 1:2. JudgeFlow obtains a clear improvement over AFlow on this harder benchmark (44.7% vs. 42.0%), confirming that our method provides effective optimization signals leading to more efficient optimization in more challenging tasks.
>
> ---
>
> [1] Zhang, Jiayi, et al. "AFlow: Automating Agentic Workflow Generation." The Thirteenth International Conference on Learning Representations (2025).
>
> [2] Zheng, Chengqi, et al. "MermaidFlow: Redefining Agentic Workflow Generation via Safety-Constrained Evolutionary Programming." arXiv preprint arXiv:2505.22967 (2025).

---

> > ### Comment · Reviewer_jeni · 2025-11-28
> >
> > Thank you for the responses. The authors have addressed most of my concerns, and I am happy to raise my score.

---

### Official Review · Reviewer_xD9d · 2025-11-01

**Soundness:** 3
**Presentation:** 3
**Contribution:** 2
**Rating:** 4
**Confidence:** 3

**Summary:**

This paper proposes **JUDGEFLOW**, an *Evaluation–Judge–Optimization–Update* pipeline for automated agentic workflow optimization. The system abstracts workflows into logic blocks (sequence/loop/condition), runs them on tasks, and applies an LLM-based Judge to assign responsibility scores to blocks in failed executions. The optimizer then modifies the weakest block via add/modify/delete operations. Experiments on GSM8K, MATH, MBPP, and HumanEval show consistent improvements over strong baselines, including AFlow and MermaidFlow, with ablations demonstrating the value of the judge and block abstraction.

**Strengths:**

1. Introduces block-level credit assignment for workflow refinement, moving beyond global end-to-end signals.

2. Experimental setup is clear and robust: multiple reasoning and coding benchmarks, consistent evaluation criteria, and ablations on logic blocks and judge components.

3. Well-written and easy to follow; modular structure and pseudocode clarify pipeline design. Case studies make the pipeline behavior interpretable.

4. Offers a more granular optimization signal for agentic workflows, improving interpretability and sample efficiency.

**Weaknesses:**

1. **Incremental novelty over AFlow**

The pipeline structure closely follows AFlow, raising concerns about incremental contribution. The primary change is shifting judgment from operator-level to block-level, while reusing the same operator abstraction and workflow-editing paradigm. The paper does not explain why this granularity shift should fundamentally improve workflow optimization, and no theoretical justification is provided.

Furthermore, the optimization remains benchmark-specific, without evidence of query-level generalization or transferable workflow patterns across tasks. As a result, the gains appear tied to engineering-level refinements rather than a general workflow-learning mechanism, limiting the conceptual novelty and broader research contribution.


2. **Limited cost–benefit analysis**

The paper does not provide a clear cost analysis. The method uses a similar amount of training data as prior work without reducing sample requirements, making it unclear whether the approach offers a meaningful efficiency improvement.

**Questions:**

1. How does the method scale with larger workflows (M≫3)? Does block-level scoring remain reliable as depth increases?

2. Is the system robust to noisy judge outputs? Any safeguards (e.g., majority voting, statistical filtering)?

3. Can the method reduce training cost compared to AFlow while maintaining performance? That would strengthen claims of efficiency.

4. Are there empirical results showing cross-benchmark generalization?

---

> ### Author Response · Authors · 2025-11-27
>
> We sincerely thank the reviewer for the time and insightful feedback. We addressed your concerns as follows.
>
>
> **[W1] Clarification of Novelty**
>
> We respectfully clarify that the Logic Blocks are not merely a simple shift, but an abstraction designed to enable our Judge mechanism. We provide the theoretical justification for this design choice below:
>
> 1. **Many existing code-presented workflow optimization lacks structural interpretability.** While code-presented workflows can offer a vast search space, optimizing these workflows is still challenging due to the lack of structural interpretability. Previous end-to-end methods like AFlow only provide the final "binary" feedback for each evaluation, leading to the absence of directional guidance within the workflow.
> 2. **Operator-level attribution is unstable due to dynamic control flow.** In code-presented workflows, operators often interact through complex control flows (e.g., if-else branches), leading to attributing blame to a single operator being difficult because execution paths vary dynamically per query (e.g., an operator in an unexecuted else branch), potentially leading to noisy and inconsistent diagnostic signals across different data samples.
> 3. **Logic Blocks serve as the stable semantic unit for the Judge.** By introducing the Logic Blocks, we encapsulate these dynamic control flows into a stable semantic unit, where Judge is able to assign more stable attribution without the ambiguity caused by different execution paths.
>
> **[W2] Cost Analysis**
>
> Our experiments show that JudgeFlow offers superior efficiency regarding optimization convergence. As shown in Figure 4b, JudgeFlow reaches high performance within the first 5 iterations, whereas AFlow requires significantly more iterations to improve. While the Judge module adds inference calls during the evaluation phase, the reduction in the number of optimization rounds offsets this. Compared with a 'blind' optimizer wasting resources, JudgeFlow has a more advanced and clearer target, making it more efficient.
>
> **[Q1] About Scaling of the Workflow**
>
> We thank you for this question. In our experiments, we set $M$ at most 3, based on the benchmarks and the capabilities of the backbone LLM, and achieved better performance. At the same time, for more complex tasks requiring deeper workflows, as workflow depth increases, the search space grows exponentially and the optimization becomes harder, similarly to the issues of sparse rewards in reinforcement learning. However, our JudgeFlow could address this by providing block-level diagnostic signals, making the optimization of the deep workflows more tractable and potentially more efficient.
>
> **[Q2] About Judge Robustness**
>
> Our JudgeFlow explicitly safeguards through statistical filtering. Instead of a single potentially noisy judge output, JudgeFlow collects traces from all failed instances and aggregates responsibility scores using the `OverallWorst` mechanism. We follow the recent finding in [1]; while individual LLM-based failure attribution might contain noise, the aggregated distribution of responsible logic blocks is more consistent with the "ground truth" of the Judge.
>
> **[Q3] Additional Cost Analysis**
>
> Besides Figure 5b, we analyze the token cost in our JudgeFlow. While JudgeFlow introduces an additional Judge module for LLM calls, our analysis reveals that the dominant cost in agentic workflow optimization lies in the Evaluation phase rather than the Judge module in our proposed methods. For example, we monitor the token consumption and cost (`gpt-4o-mini`) for a single optimization round on the GSM8K dataset, and the Evaluation costs \\$0.45 and the Judge costs \\$0.01, the ratio of Judge/Evaluation is only 2%.
>
> **[Q4] Cross Benchmark Generalization**
>
> To demonstrate the generalization of JudgeFlow in the revised manuscript, we test the cross-benchmark transfer, including the Math transfer and Code transfer. We optimize the workflow based on the MATH(MBPP) dataset and zero-shot evaluation on GSM8K(HumanEval) dataset. We report the solve rate (%) on GSM8K and pass@1 on HumanEval as evaluation metrics as follows.
>
> | Optimization → Evaluation | MATH→GSM8K | MBPP→HumanEval |
> | :------------------------ | :---------- | :-------------- |
> | AFlow[2]                  | 91.95       | 90.84           |
> | JudgeFlow                 | 92.89       | 93.89           |
>
> ---
>
> [1] Zhang, Shaokun, et al. "Which Agent Causes Task Failures and When? On Automated Failure Attribution of LLM Multi-Agent Systems." Forty-second International Conference on Machine Learning (2025).
>
> [2] Zhang, Jiayi, et al. "AFlow: Automating Agentic Workflow Generation." The Thirteenth International Conference on Learning Representations (2025).

---

### Official Review · Reviewer_XB7y · 2025-11-03

**Soundness:** 2
**Presentation:** 4
**Contribution:** 2
**Rating:** 2
**Confidence:** 2

**Summary:**

Previous work in agentic workflow optimization agents rely on a three stage system (Evaluation, Optimization, Update) to optimize an agentic workflow. One key issue with this approach is that when multiple updates are needed, its hard for the optimization layer to prioritize the most problematic block. Consecutively,  the authors hypothesize that introducing an additional self-reflection layer, namely a Judge, after the evaluation stage but before the optimization layer can inform the decision of the LLM optimizer. Experiments on four mathematical reasoning and code generation domains show superior performance.

**Strengths:**

The paper is very well written and I commend the authors for putting in the extra work for including psuedocode and useful figures.

**Weaknesses:**

**Baselines:** I'm not convinced that the current results demonstrate the effectiveness of the method properly:
 - HumanEval and MBPP are widely regarded as saturated in code generation, partly because most frontier models have been trained on these datasets. Please show additional results on LiveCodeBench, AIME2025, SWE-Bench-Live, etc. which have more complex tasks.
 - The set of baselines do not seem reflective of the current state of the art either. How does this compare with OpenHands, SWE-Agent or other well-established agents' performance on the code generation benchmarks listed above?
 - For example, `o1-mini` archives `96.2%` pass@1 on HumanEval (which isn't mentioned in this paper). The datasets mentioned in this paper might be of interest as well as our understanding of how to evaluate methods on code geenration tasks has evolved [https://arxiv.org/pdf/2412.21199](https://arxiv.org/pdf/2412.21199).

**Overall:** I'm in favor of rejecting this work because I'm not sure if the current set of baselines and tasks is reflective of the best methods for code generation in the community. I urge the authors to find more competitive baselines as well as more competitive datasets to evaluate JudgeFlow.

**Questions:**

See Weaknesses.

---

> ### Author Response · Authors · 2025-11-27
>
> Thanks for your valuable feedback. Regarding the scope of our paper, we would like to offer the following clarifications.
>
> **Additional Benchmark Results**
>
> Following previous works such as AFlow[1] and MermaidFlow[2], we primarily used HumanEval and MBPP as our benchmarks to ensure a fair comparison.  However, we agree with the reviewer that harder benchmarks are important. On AIME, which is substantially more challenging than GSM8K/MATH, JudgeFlow improves the performance over AFlow. The results are as follows:
>
>
> | Method     | Result (Mean ± Std Dev) |
> |------------|--------------------------|
> | AFlow      | 0.4200 ± 0.0460          |
> | JudgeFlow  | 0.4467 ± 0.0278          |
>
>
> In the additional experiments in the revised manuscript, we directly utilized the official implementation of AFlow and our implementation of JudgeFlow. To balance performance and cost for this harder benchmark, different from `gpt-4o-mini` used in our original paper, we used `gpt-4_1-mini-2025-04-14` as both Judge, Optimization, and Execution LLM backbone. Following the previous work and our paper, we set the temperature to 0. We sampled the validation data from the AIME 2024 dataset during optimization and evaluated the performance on the AIME 2025 dataset. Given the limited sample size of AIME, we established a validation-to-test ratio of 1:2. JudgeFlow obtains a clear improvement over AFlow on this harder benchmark (44.7% vs. 42.0%), confirming that our method provides effective optimization signals leading to more efficient optimization in more challenging tasks.
>
> **About Baseline Selection**
>
> While OpenHands and SWE-Agent have achieved great success in code generation, we respectfully clarify that our contribution is different in scope. Unlike the previous hand-crafted and highly specialized systems, JudgeFlow is a general automated agentic workflow optimization framework across diverse domains. In our experiments, we used code generation and mathematical reasoning tasks and mainly focused on demonstrating optimization efficiency rather than making a domain-specific agent, like a coding agent.
>
> **About `o1-mini` and HumanEval**
>
> We acknowledge the impressive performance that `o1-mini` achieves 96.2% pass@1 on HumanEval. However, we respectfully emphasize that `o1-mini` represents a stronger backbone LLM, while our JudgeFlow is a workflow optimization framework. To evaluate the performance compared with previous works fairly, we strictly controlled the backbone LLM (`gpt-4o-mini` following MaAS[3] and MermaidFlow[2]). At the same time, our framework is not tied to a specific model. As demonstrated in Table 2, we also evaluated the effectiveness of JudgeFlow using different LLMs as judge and optimization backbone LLMs (including `gpt-4o` and `gemini-2.5-flash`, which are more expensive models). Regarding the datasets and the evaluation methods, since our paper aims to improve the performance of a given (potentially weak) backbone LLM, we think that the standard datasets and evaluation methods used in the previous work remain valid and appropriate.
>
> ---
>
> [1] Zhang, Jiayi, et al. "AFlow: Automating Agentic Workflow Generation." The Thirteenth International Conference on Learning Representations (2025).
>
> [2] Zheng, Chengqi, et al. "MermaidFlow: Redefining Agentic Workflow Generation via Safety-Constrained Evolutionary Programming." arXiv preprint arXiv:2505.22967 (2025).
>
> [3] Zhang, Guibin, et al. "Multi-agent Architecture Search via Agentic Supernet." Forty-second International Conference on Machine Learning (2025).

---

### Author Response · Authors · 2025-12-03

We thank all reviewers for their tremendous efforts in reviewing our paper and providing valuable feedback. We appreciate the constructive comments and have carefully addressed each concern during the rebuttal period.

We are pleased that reviewers acknowledge the key strengths of our work. Specifically, reviewers recognized that our paper identifies an important gap in existing workflow optimization methods that rely solely on end-to-end evaluation signals (`xD9d`, `jeni`, `hjhQ`), that the experimental setup is clear and robust with comprehensive ablation studies (`xD9d`, `jeni`), that the paper is well-written with clear presentation and good organization (`XB7y`, `xD9d`, `jeni`, `hjhQ`), and that the Judge-guided optimization mechanism significantly enhances both efficiency and effectiveness of workflow optimization (`jeni`, `hjhQ`).

The major concerns raised by reviewers and our responses are summarized as follows:

1. Benchmark and baseline selection (`XB7y`, `jeni`). Following previous works such as AFlow and MermaidFlow, we primarily used standard benchmarks to ensure fair comparison. However, we acknowledge the importance of harder benchmarks and conducted additional experiments on AIME 2025, a substantially more challenging benchmark than GSM8K/MATH. JudgeFlow achieves 44.67% accuracy compared to AFlow's 42.00%, demonstrating effective optimization on complex reasoning tasks. Regarding baselines, we clarify that our contribution is a general automated agentic workflow optimization framework rather than a domain-specific coding agent, which differs from systems like OpenHands and SWE-Agent.

2. Novelty concerns (`xD9d`, `jeni`, `hjhQ`). We clarify that Logic Blocks are not merely a simple abstraction shift but are specifically designed to enable our Judge mechanism. Code-presented workflows lack structural interpretability, and operator-level attribution is unstable due to dynamic control flows. Logic Blocks encapsulate dynamic control flows into stable semantic units, allowing the Judge to assign responsibility without ambiguity caused by varying execution paths. The ablation study in Table 3 confirms the importance of both Logic Blocks and the Judge module.

3. Cost and efficiency analysis (`xD9d`, `jeni`, `hjhQ`). Our analysis reveals that the Judge module introduces minimal overhead. While JudgeFlow introduces an additional Judge module for LLM calls, the dominant cost in agentic workflow optimization lies in the Evaluation phase rather than the Judge module in our proposed methods. Furthermore, as shown in Figure 5b, JudgeFlow converges significantly faster than baselines, reaching high performance within the first five iterations, which offsets the additional Judge overhead.

4. Judge robustness and validation (`hjhQ`). JudgeFlow employs statistical filtering through the `OverallWors` mechanism, aggregating responsibility scores across all failed instances rather than relying on single potentially noisy outputs. While explicit ground-truth attribution labels are unavailable, the optimization outcomes indirectly validate that the Judge identifies meaningful failure sources. Additionally, we conducted cross-benchmark transfer experiments showing JudgeFlow achieves state-of-the-art results when optimizing on MATH/MBPP and evaluating on GSM8K/HumanEval.

Following reviewer suggestions, we have updated the manuscript to include additional AIME 2025 experiments, cross-benchmark generalization results, detailed cost analysis, and clarification on the theoretical justification for Logic Blocks. We thank the reviewer `jeni` expressed the willingness to raise their score after our response addressed their concerns. We believe JudgeFlow provides a valuable contribution to the community by offering fine-grained diagnostic signals for agentic workflow optimization, improving both interpretability and sample efficiency.

---

### Meta-Review · Area_Chair_CZzW · 2026-01-04

**Summary:**

Across reviews, the main concerns that informed the decision are concrete and largely shared. Reviewers questioned whether JudgeFlow represents a sufficiently novel advance beyond prior agentic workflow optimization methods such as AFlow, as the core pipeline and edit operations remain largely unchanged and the main contribution is a finer attribution granularity. There were also concerns that the original evaluation relied on saturated or outdated benchmarks and non-competitive baselines, especially for code generation, limiting the strength of the empirical evidence. Finally, reviewers raised questions about the practical benefits of the Judge module, including cost efficiency, robustness of block-level attribution without ground-truth validation, and whether the reported gains justify the added complexity.

**Reviewer Concerns:**

1. Reviewer XB7y took a clearly negative stance. The reviewer argued that HumanEval and MBPP are saturated and not representative of current code generation evaluation, and that comparisons against modern systems such as OpenHands or SWE-Agent were missing. While the rebuttal added AIME 2025 results and clarified that JudgeFlow targets general workflow optimization rather than domain-specific agents, this did not address the reviewer’s core concern about relevance to state-of-the-art benchmarks and baselines. This concern remains outstanding.

2. Reviewer xD9d viewed the work as technically solid but incrementally novel. The reviewer questioned whether moving from operator-level to block-level attribution fundamentally changes workflow optimization, and initially noted missing cost analysis, robustness discussion, and cross-benchmark generalization. The rebuttal addressed most of these points by providing theoretical motivation for logic blocks, explicit token-level cost analysis, aggregation-based robustness mechanisms, and cross-benchmark transfer experiments. However, the concern that the contribution is primarily an engineering refinement rather than a new conceptual framework is only partially resolved.

3. Reviewer jeni was initially marginally negative, citing incremental novelty, computational cost, and outdated benchmarks. The rebuttal directly responded with additional AIME experiments, efficiency analysis, and clarification of the judge-guided optimization benefits. This reviewer explicitly stated they were happy to raise their score, suggesting their concerns were largely addressed.

4. Reviewer hjhQ expressed mixed views. While acknowledging clear motivation, good presentation, and consistent empirical gains, the reviewer questioned the novelty of using LLM-as-a-judge with simple rank-based scoring, the lack of ground-truth validation for attribution, and the absence of cost analysis. The rebuttal provided concrete cost numbers and argued for indirect validation via optimization outcomes and aggregated judging. These responses mitigate but do not fully eliminate concerns about technical depth and attribution validity.

**Reviewer Scores:**

1. XB7y: Likely unchanged and still negative, as the rebuttal did not convincingly address concerns about benchmark relevance and baseline competitiveness.
2. xD9d: Likely unchanged or only slightly improved, remaining borderline due to persistent concerns about incremental novelty despite stronger analysis.
3. jeni: Likely improved from marginal reject to borderline accept, as explicitly indicated after the rebuttal.
4. hjhQ: Likely unchanged, remaining borderline with continued reservations about novelty and attribution methodology.

---

### Decision · Program_Chairs · 2026-01-26

Reject